# Comparative Analysis of Force and Eddy Current Position Sensing Approaches for a Magnetic Levitation Platform with an Exceptional Hovering Distance

Reto Bonetti [1], Spasoje Mirić [2,*] and Johann W. Kolar [1,*]

1    Power Electronic Systems Laboratory, ETH Zurich, 8092 Zürich, Switzerland; bonetti@lem.ee.ethz.ch
2    Innsbruck Drive and Energy Systems Laboratory, University of Innsbruck, 6020 Innsbruck, Austria
*    Correspondence: spasoje.miric@uibk.ac.at (S.M.); kolar@lem.ee.ethz.ch (J.W.K.); Tel.: +43-512-507-62780 (S.M.)

**Abstract:** This paper provides a comparative analysis between a force sensor and an eddy current sensor, focusing on their usability to determine the position of a circular levitating permanent magnet (PM) mover within an axially symmetric magnetic levitation platform (MLP) with an exceptionally large air gap. The sensors enable closed-loop control, which is essential for accurately and stably maintaining the mover's radial position. For the considered MLP, a change in radial position in principle results in a tilting of the mover, i.e., a deviation from the parallel alignment relative to the stator. As both the radial position and the tilting angle affect the sensors' (force and eddy current) output voltage, an observer must deduce the radial position from the output sensor's voltage, requiring a comprehensive MLP dynamic model and calibration of the models for both sensing approaches. The paper discusses the advantages and weaknesses of each sensor concept, exploring operational principles and performance in levitation tests. The force sensor exhibits versatility, proving functional across various application scenarios, such as when the mover is sealed in a conductive, non-magnetic chamber. In contrast, due to its high-frequency operation, the eddy current sensor is more straightforward to characterize, simplifying its behavior relative to the mover's slower dynamics. Measurements are conducted to validate the models, showing the eddy current sensor's robustness against disturbances and imperfections in the MLPs and its immunity to cross-axis interference. Conclusively, in levitation experiments where the mover is vertically distanced at 104 mm from the stator, the eddy current sensor achieves a position tracking precision about ten times better and a signal-to-noise ratio (SNR) ten times higher compared to the off-the-shelf force sensor, confirming its better performance and reliability; however, it cannot be used in applications where conductive objects are present in the air gap. Furthermore, additional experiments are conducted on the MLP using the eddy current sensor to show the controller's robustness and dynamic reference tracking capability, with and without a payload.

**Keywords:** magnetic levitation platform; force sensor; eddy current sensor; dynamic model



## 1. Introduction

Magnetic levitation finds applications in various fields, such as transportation, aerospace, civil, biomedical, chemical, architectural, and automotive engineering, as detailed in [1]. Within this extensive range of applications, magnetic levitation platforms (MLPs), which are characterized by the absence of mechanical contact between the levitated platform (mover) and the steady base (stator), serve as vibration isolation systems for high-precision manufacturing [2,3], dynamic supporting structures for mirrors in optical pointing and scanning applications [4], zero-power gravity compensation systems [5], e.g., nanometer-scale positioning [6], ground-based testing of large optical equipment for space use [7], and cleanroom conveyor systems [8]. Magnetic levitation and precise position measurement are also important when using highly sensitive temperature-compensated quartz sensors that measure position based on capacitive change, as shown in [9,10].

The theoretical and practical implementation of an axially symmetric MLP potentially suitable for zero-power gravity compensation applications was analyzed in [11] and [12] with the aim of extending the vertical air gap (i.e., the levitation height $h$ in Figure 1) between the stator and mover. Similar systems based on commercially available levitation modules have been sporadically analyzed in the literature regarding structural optimizations and closed-loop control [13–18].

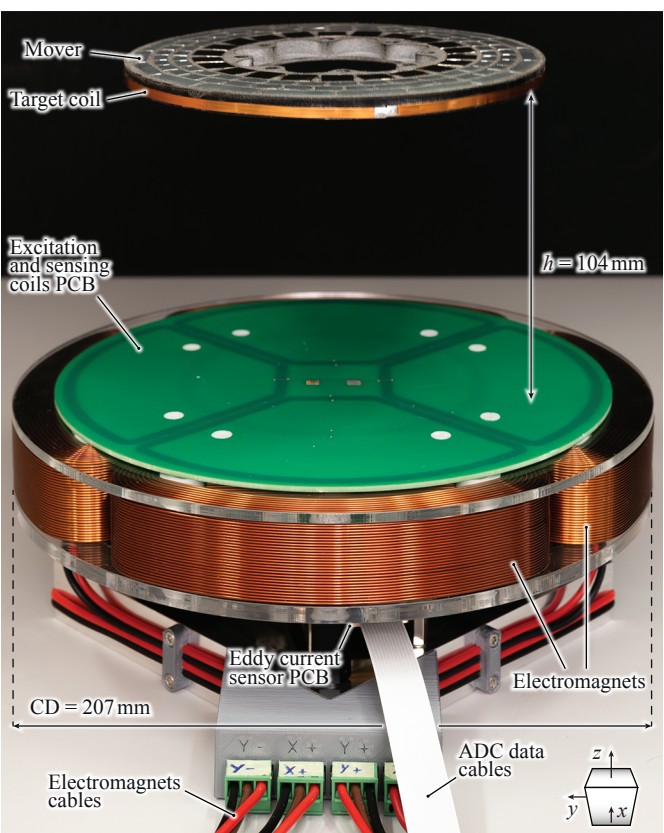

**Figure 1.** Magnetic levitation platform (MLP) with the mover levitating using the eddy current position sensor (ECS) for determining the radial ($x, y$-axis) position. The permanent magnet (PM) stator, which is not visible as it is covered by the excitation and sensing PCB coils, generates axial, i.e., $z$ direction, magnetic forces that maintain the mover at the reported distance of $h = 104$ mm. The electromagnets compensate for magnetic destabilizing forces in the radial $x, y$ plane and maintain the mover at the desired position based on continuous measurements captured by the position sensor.

Such MLPs are formed by a hybrid structure of permanent magnets (PMs) and electromagnets (EMs), which constitute the passive and the active part of the system, respectively. Stator and mover PMs generate the vertical force that compensates for gravity. EMs only generate the stabilizing forces that hold the mover at the zero-power point, i.e., the radially centered position $x = y = 0$, resulting in lower volume and losses. The PMs are designed so that the mover has three passively stable degrees of freedom (DOFs), achievable with the axially symmetric geometry presented (see Figure 1). Specifically, the axial ($z$ direction) displacement and the two rotations around the radial axes $x, y$ are stable due to restoring magnetic forces and torques. The radial ($x, y$ direction) displacement is passively unstable due to magnetic forces that pull the mover away from the center $x = y = 0$. Therefore, an active control of these two DOFs is strictly necessary. The axial symmetry of both PMs contributes to marginal stability in rotation around the $z$-axis. This means there is no strict need for active control to manage this rotation; therefore, this paper does not address it.

The adoption of closed-loop control is essential to stabilize the mover's unstable radial positions, denoted by $x_{\mathrm{m}}$ and $y_{\mathrm{m}}$. This necessitates an effective method for obtaining the

position of the MLP's mover, especially in scenarios demanding high versatility, such as, e.g., operation within stainless-steel (non-magnetic) process chambers where traditional optical position sensing is infeasible. Consequently, [12] introduces a novel position measurement methodology that relies on observing the reaction forces exerted on the stator by the mover's movements, called the *reaction force-based* position sensor (RFS). This approach is notably beneficial for facilitating the automated manufacture and manipulation of objects, enabling robot arms to seamlessly navigate through the air gap between the stator and the mover. However, the accurate implementation of an RFS necessitates meticulous modeling of the MLP's dynamics to convert reaction forces into position data precisely. While the complexity and noise inherent in the force sensor may impact the precision of the RFS method, the technique's universal applicability and ability to accommodate conductive objects within the air gap stand out. The second advantage could also be achieved by Hall effect sensors, which are, however, not applicable to this MLP because the high magnetic field near the stator PM would saturate the measuring circuit output [12]. The considerable advantage of accommodating conductive materials in the air gap is not achievable with other types of position sensors, such as eddy current-based sensors (ECSs). Hence, the objective of this paper is to meticulously evaluate the RFS and ECS technologies, focusing on their applicability, the complexity of implementation and modeling, accuracy, stability, and reference tracking capabilities. Such a detailed comparison is critical when selecting the most appropriate sensor type for specific applications, easing decision-making in the integration process of advanced mechatronics.

In Section 2, we introduce the geometry and display the operating principles of the ECS. A brief repetition of the operating principles of the RFS, which are extensively discussed in [12], is given in Section 3. The position sensor's dynamics modeling and verification is covered in Section 4. The comparative results of steady-state levitation experiments for both sensors are presented in Section 5, where we clarify that there are more restrictions in the design of an RFS-based position control, such as a more sophisticated dynamical model requirement and lower signal-to-noise ratio, compared to the ECS. However, the ECS cannot be used in applications when the mover is enclosed in a conductive chamber due to Faraday's cage effect. Furthermore, in Section 6, additional tests on the MLP using only the ECS demonstrate that dynamical reference tracking and loading the mover with extra weight are possible. Finally, conclusions summarizing the main findings are drawn in Section 7.

## 2. Eddy Current Position Sensor for the MLPs

This section first introduces the ECS, tailored explicitly for the analyzed MLP. Next, the main properties of the RFS concept proposed in [12] are summarized in Section 3, and a detailed comparison of the two position-sensing technologies is finally performed starting from Section 4.

### 2.1. ECS Geometry and Operating Principle

The ECS applied to the MLP employs high-frequency electromagnetic signals to determine the position of the mover in space. A set of four coils is used to sense the radial $x, y$ position and is arranged as depicted in Figure 2a,b. The excitation coil is radially centered and located directly above the stator (see Figure 2a). A time-varying voltage $u_{\mathrm{exc,ECS}}$ with a high-frequency $f_{\mathrm{exc}}$ is applied at its terminals (see Figure 2c), whose current generates a high-frequency magnetic field that reaches and couples with the target coil on the mover. The mover is equipped with a short-circuited target coil, where an eddy current flows to counteract the coupled excitation flux (Faraday's law of induction), generating a magnetic field with the same frequency as the excitation frequency (denoted as "ECS target flux" in Figure 2a).

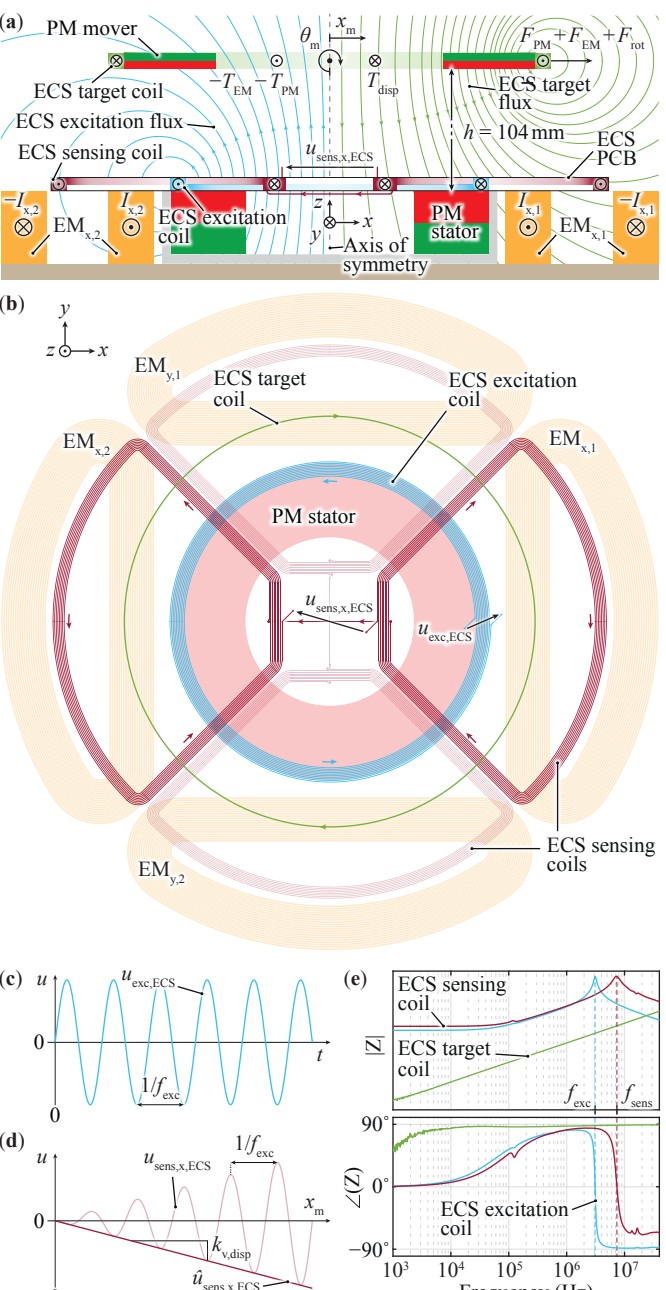

**Figure 2.** (**a**) Section view of the MLP that uses the ECS, where the direction of the magnetic flux density, induced currents, and voltages is shown for a positive excitation current. The directions of the shown electromagnetic force $F_{EM}$ and torque $T_{EM}$ are given by a positive current injected in the EMs when the mover is at its natural levitation point $x_m = 0$ and $\theta_m = 0$. Additionally, the effects of displacements $x_m$ and $\theta_m$ are shown, leading to the displacement-dependent magnetic force $F_{PM}$ (unstable) and torque $T_{disp}$, and the rotation-dependent magnetic torque $T_{PM}$ (stable) and force $F_{rot}$ [11]. (**b**) Top view of the system (excluding the radially centered mover) that shows the spatial distribution of the sensor's coils with respect to the stator and EMs. (**c**) High-frequency excitation voltage for the excitation coil. (**d**) Expected voltage output from the sensing coil $u_{sens,ECS}$ and demodulated voltage signal $\hat{u}_{sens,ECS}$ for a positive mover's displacement in the $x$ direction. (**e**) Impedance measurements of the proposed sensor's coils, where the target and sensing coils operate in the inductive region at the excitation frequency $f_{exc} = 3.1\,\text{MHz}$, indicating that magnetic coupling exists.

Two sensing coils per axis are connected in anti-series to couple the flux from the excitation and target coil, generating a proportional voltage $u_{\text{sens,ECS}}$ at their open terminals. Examining the sensing coils (e.g., the one for position measurement along the $x$-axis) closely, the voltage generated by the excitation coil's magnetic field is zero due to the anti-series connection of the sensing coils and the equal coupling of excitation flux (a radially symmetric magnetic field does not induce any sensing voltage). The same applies to the field from the target coil when the mover is radially centered and horizontal. As the mover displaces from the center (e.g., in the positive $x$ direction), the $x$-axis sensing coil outputs a time-varying voltage with frequency $f_{\text{exc}}$ as illustrated in Figure 2d because of an imbalance in the coupled target flux between the two sides of the coil. When the sensed signal is demodulated to obtain the amplitude $\hat{u}_{\text{sens,ECS}}$, a linear relation between the mover's displacement and sensed voltage valid for small displacements is observed, i.e., $\hat{u}_{\text{sens,ECS}} = -k_{\text{v,disp}} \cdot x_{\text{m}}$, where $k_{\text{v,disp}}$ is the sensitivity for the radial displacement and is given in Table 1. As already observed in [12] for the reaction force sensor, the tilting of the mover around the radial $x, y$-axes influences the sensed voltage. Tilting the mover while keeping it radially centered, a linear relation valid for small angles is observed and written as $\hat{u}_{\text{sens,ECS}} = -k_{\text{v,rot}} \cdot \theta_{\text{m}}$, where $k_{\text{v,rot}}$ is the sensitivity for the rotation. Considering both the displacement and the rotation, which always happen simultaneously [12] during destabilization and control action, the superimposed equation

$$\hat{u}_{\text{sens,ECS}} = -k_{\text{v,disp}} \cdot x_{\text{m}} - k_{\text{v,rot}} \cdot \theta_{\text{m}} \tag{1}$$

fully describes the mover's position in space for small displacements and angles.

**Table 1.** ECS parameters, where the values of the *RLC* components are extracted from the impedance measurement depicted in Figure 2e performed with the coils installed on the MLP. Due to symmetry, only the $x$-axis sensing coil parameters are shown.

| | | | |
|---|---:|---|---:|
| $k_{\text{v,disp}}$ | 51.3 µV/mm | $k_{\text{v,rot}}$ | 5 µV/° |
| $k_{\text{IPS}}$ | 240 V/V | $T_{\text{f,ECS}}$ | 4.6 ms |
| $f_{\text{f,ECS}}$ | 35 Hz | $u_{\text{exc,ECS}}$ | 7.4 $V_{\text{pp}}$ |
| $f_{\text{exc}}$ | 3.1 MHz | $f_{\text{sens}}$ | 7.6 MHz |
| $R_{\text{exc}}$ | 9.5 Ω | $L_{\text{exc}}$ | 23.9 µH |
| $R_{\text{sens}}$ | 15 Ω | $L_{\text{sens}}$ | 27.2 µH |
| $R_{\text{target}}$ | 2 mΩ | $L_{\text{target}}$ | 0.4 µH |
| $C_{\text{exc}}$ | 110 pF | $C_{\text{sens}}$ | 16 pF |
| $N_{\text{exc}}$ | 10 | $N_{\text{sens}}$ | 16 |

*2.2. ECS Design Considerations*

For the design of the ECS coils, the spatial distribution and the number of turns are important metrics because they primarily determine the coils' inductance and parasitic capacitance, ultimately defining the resonant frequency. The goal is to maximize the sensitivity $k_{\text{v,disp}}$ and keep the inductive behavior and the overall MLP's size the same in order to facilitate a fair comparison with the RFS. The outermost winding of the excitation coil is chosen to be as large as possible to maximize coupling with the target coil on the mover. However, the size is limited by the placement of the PM stator and EMs because the excitation coil's magnetic field induces eddy currents in neighboring conductive materials, affecting the coupling with the target coil. The distribution of eddy currents in a conductive material is determined by the skin depth $\delta = \sqrt{\rho / \pi f_{\text{exc}} \mu_0 \mu_{\text{r}}}$, which depends on its electrical resistivity $\rho$ and relative permeability $\mu_{\text{r}}$ [19]. Copper EMs have a much lower resistivity than NdFeB PMs ($\rho_{\text{Cu}} = 1.68 \times 10^{-8} \, \Omega\text{m} \approx 0.12 \cdot \rho_{\text{NdFeB}}$) and a similar relative permeability ($\mu_{\text{r,Cu}} \approx 1 \approx 0.93 \cdot \mu_{\text{r,NdFeB}}$). Consequently, the skin depth in EMs is smaller than in PMs $\delta_{\text{Cu}} \approx 37 \, \mu\text{m} \approx 0.36 \cdot \delta_{\text{NdFeB}}$, indicating a higher eddy current density on the material's surface near the excitation coil. Therefore, the detrimental field caused by eddy currents in EMs in the vertical space above the excitation coil is more pronounced than the one resulting from eddy currents in the stator PMs. Consequently,

when limiting the excitation coil's diameter to the dimension of the stator PM, a larger coupling with the target coil is expected, even though the coil's diameter is smaller than the MLP's characteristic dimension CD. The larger coupling between the excitation and the target coil is beneficial as larger eddy currents are induced in the target coil, finally leading to larger induced voltage in the sensing coils and, thus, a better sensitivity. The same principle holds for the sensing coils, where the area of the outermost winding is maximized while minimizing the overlapping region with the stator and, most importantly, with the EMs (see Figure 2b). A total diameter of the sensing coils larger than the MLP's characteristic dimension is disregarded to preserve compactness. Regarding the target coil, a single-turn coil is applied on the mover's lateral surface so that the total weight and outer diameter are only marginally increased while maximizing the coupling area for the excitation coil's magnetic field. Further, the integrated circuit (IPS2550 [20]) used to drive the excitation coil, and to amplify and demodulate the high-frequency signals from the sensing coils, poses an additional constraint on the coils' design. Specifically, the excitation coil has to exhibit a resonant frequency in the frequency range $f_{exc} = [2\,\mathrm{MHz}, 5.6\,\mathrm{MHz}]$ to be appropriately excited. A high number of turns $N$ and an operating frequency $f_{exc}$ close to the maximum are advantageous for achieving a large sensitivity since the induced voltage in the sensing coils directly depends on both design parameters (Faraday's law of induction). However, the coils' parasitic capacitances $C_{exc}$ and $C_{sens}$ limit the number of turns since they increase for greater $N_{exc}$ and $N_{sens}$. The proposed design features a PCB carrying the excitation coil with $N_{exc} = 10$ turns and the sensing coils with $N_{sens} = 16$ turns, which are arranged as depicted in Figure 2b. The measured impedance of all the eddy current sensor coils is given in Figure 2e, where the operating frequency is $f_{exc} = 3.1\,\mathrm{MHz}$, and the sensing coils' resonant frequency is $f_{sens} = 7.6\,\mathrm{MHz}$. The numerical values of the equivalent lumped elements are listed in Table 1.

The measurement bandwidth of the eddy current sensor is limited by the IPS2550's internal demodulation process, which removes the high-frequency carrier signal to obtain low-frequency demodulated signals that depend on the mover's position. This process introduces a constant time delay of $T_d = 4\,\mu s$ valid in the whole operating frequency range. In the frequency domain, the time delay is approximated by a first-order transfer function (TF) $H_d(s) = (2 - T_d s)/(2 + T_d s)$ [21] with a zero and a pole at $f_d = 1/(\pi \cdot T_d) = 79.6\,\mathrm{kHz}$. Accordingly, the sensor's bandwidth is defined as the $-3\,\mathrm{dB}$ frequency of a phase-equivalent TF $\widetilde{H}_d(s) = 1/(2 + T_d s)^2$ and lies at $f_{bw,ECS} = f_d \cdot \sqrt{\sqrt{2} - 1} = 51.2\,\mathrm{kHz}$. Since the approximate bandwidth is much larger than the mover's dynamics (maximum $f_{n,rot} = 2\,\mathrm{Hz}$ for the tilting as reported in Table 3 and [12]), we neglect the dynamics of the ECS during the MLP dynamics analysis and the position controller design.

## 2.3. Impact of the Conductive Obstacles in the Air Gap

As previously highlighted, the RFS offers a broader range of applications than ECS, particularly due to its capability to function effectively in environments with conductive materials within the air gap. In this subsection, we examine how metal objects (aluminum and copper) influence the performance of the ECS.

Considering Figure 3, when the mover is manually displaced by 1 mm back and forth from the center without materials in the air gap, the sensed voltage of the eddy current sensor varies between zero and approximately 12.3 mV (blue curve). However, when a conductive sheet is inserted in the air gap to mimic the situation where the mover is isolated and levitated in a hermetically sealed chamber, the mover's displacement is not recognizable in the sensed voltage (see orange and yellow traces in Figure 3). This happens because the target coil on the mover is shielded from the excitation coil's magnetic field, which means that there are no eddy currents in the target coil; accordingly, no position-dependent opposing field occurs. In contrast, the RFS can still operate since the magnetic and electromagnetic forces acting on the mover are always reflected on the stator and EMs.

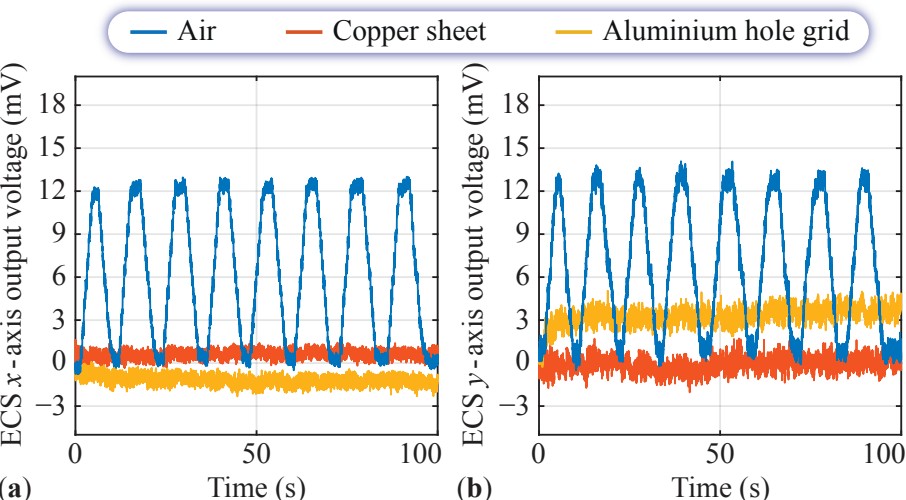

**Figure 3.** ECS's response for a manual radial displacement of the mover by 1 mm back and forth using a micropositioning stage for different mediums in the air gap. (**a**,**b**) show the results on the *x*- and *y*-axes, respectively. When a square conductive sheet (207 mm × 207 mm) that fully covers the PM stator and EMs is inserted in the middle of the air gap, i.e., at about 50 mm from the top surface of the PM stator, the magnetic coupling between the excitation, target, and sensing coils is heavily reduced due to the magnetic flux shielding, resulting in zero sensed voltage (neglecting the independent offsets, drifts, and noise). The copper sheet is 0.35 mm thick, whereas the aluminum foil is 1 mm thick and perforated, with holes having a diameter of 5 mm and a spacing of 2 mm.

## 3. Reaction Force-Based Position Sensor

This chapter briefly introduces the RFS, which is discussed in detail in [12], with the help of Figure 4. This position sensing method consists of a force sensor (Forsentek FNZ-100N [22]) capturing reaction forces on the stator caused by magnetic interactions between the PM stator and PM mover and electromagnetic interactions between the EMs mounted on the stator's baseplate and the PM mover (see Figure 4a).

Accordingly, a three-axis $(x, y, z)$ strain gauge-based force sensor captures the total reaction force $F_{RFS} = -F_{PM} - F_{EM} - F_{rot}$. Four strain gauges arranged as a Wheatstone bridge are glued to each sensing element (one per Cartesian axis $x, y, z$) and are excited with a constant voltage $u_{exc,RFS}$ (see Figure 4a,b). The total reaction force displaces the sensing side of the force sensor, resulting in the bending of the sensing element. This bending stretches the strain gauges, causing a change in their resistance, translating into a variation in the output voltage $u_{sens,RFS}$. The trend of $u_{sens,RFS}$ for a positive reaction force $F_{RFS}$ is approximated with a linear relationship, as shown in Figure 4c, according to the following relations:

1.  The movement of the force sensor's sensing side $x_s$ and the bending of the sensing element are linearly related to the applied force, i.e., $x_s = F_{RFS}/k_{s,RFS}$, where $k_{s,RFS}$ is the stiffness of the force sensor.
2.  The stretch of the strain gauges, their resistance change, and the output voltage variation are linearly dependent on the movement of the force sensor's sensing side, i.e., $u_{sens,RFS} = k_{v,RFS} \cdot x_s$, where $k_{v,RFS}$ is a constant representing the mechanical-to-electrical signal conversion.

From the sensed output voltage, information about the mover's position and tilting angle is gained since the total reaction force $F_{RFS}$ depends on the mover's radial $x, y$ position ($F_{PM}$ in Figure 4d) and the mover's tilting around the $x, y$-axes ($F_{rot}$ in Figure 4h) [12].

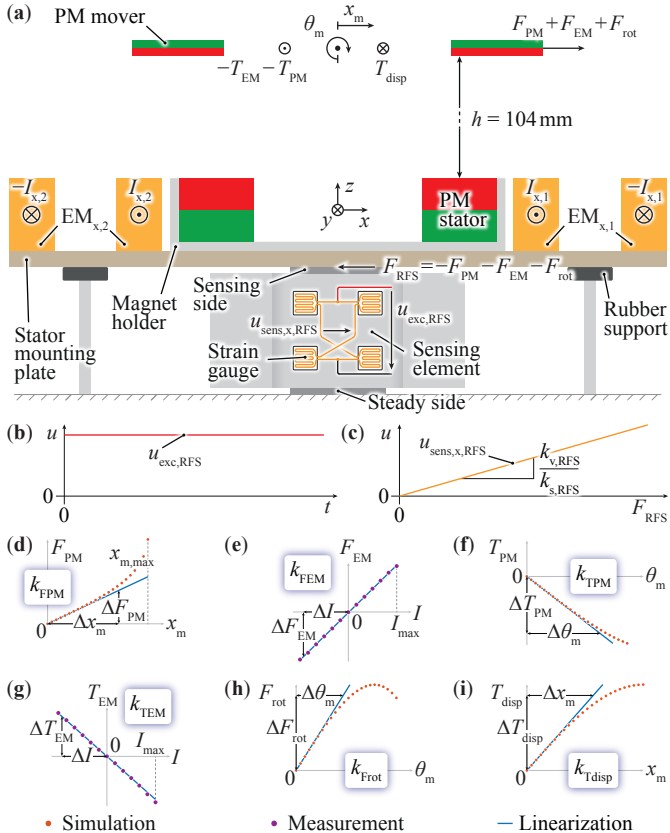

**Figure 4.** Illustrations copied from [12] and adapted. (**a**) Section view of the MLP including the RFS with the corresponding $x$-axis sensing element and strain gauges. The same arrangement is provided for measuring the $y$-axis. (**b**) Constant voltage excitation for the strain gauges forming a Wheatstone bridge. (**c**) Expected RFS voltage output for a positive reaction force $F_{\text{RFS}}$ acting on the sensing side due to the mover's motion and the electromagnetic force. (**d**–**i**) show the linearizations of different simulations and/or measurements performed on the MLP to build a model in the neighborhood of $x_{\text{m}} = 0$, $y_{\text{m}} = 0$, and $\theta_{\text{m}} = 0$.

## 4. Dynamics Modeling and Verification

Developing a dynamic model for the MLP is a prerequisite for designing a controller capable of actively managing the mover's position. As discussed in Section 2, leveraging (1) reveals that the voltage signal from the ECS contains information about the mover's radial position and tilting angle. Therefore, an observer is necessary to distill at least the mover's position from the detected voltage. Subsequently, this positional information enables the controller to guide the mover by manipulating the system's sole input, namely, the EM's currents. Moreover, due to the significant levitation height, oscillations of the tilting angle are anticipated, which are not sufficiently dampened by the passive magnetic interactions. In response, the controller needs the estimated mover's angle—deduced from the sensed voltage by the observer—to actively mitigate any ensuing oscillations. To achieve this, we introduce the linearized dynamic model of the system, which is used for such an observer. This model is validated by measurements and refined with calibration through transfer function (TF) measurements to ensure its robustness and accuracy (see [12] for more details on disturbance TFs).

### 4.1. Dynamic Model

As shown in [12] and reported here again for completeness, the mover's dynamic model is divided into two parts that describe the tilting around the radial axes $x, y$ (Figure 5a) and the radial motion (displacement from center) (Figure 5b). It should be noted that the tilting and displacement are coupled [12], which is modeled with constants

$k_{\text{Tdisp}}$ and $k_{\text{Frot}}$. For the direct comparison with the ECS, the third-order dynamic model of the force sensor used in [12] is depicted in Figure 5c. This model comprises a stable mass-spring-damper system that replicates the mechanics of the sensing element (from $F_{\text{RFS}}$ to $x_{\text{s}}$), the mechanical-to-electrical conversion of the strain gauges glued to the sensing element ($k_{\text{v,RFS}}$), the electronic amplification ($k_{\text{VGA}}$), and a first-order low-pass filter that attenuates noise and avoids aliasing effects in the subsequent analog-to-digital conversion (from $u_{\text{amp,RFS}}$ to $u_{\text{filt,RFS}}$).

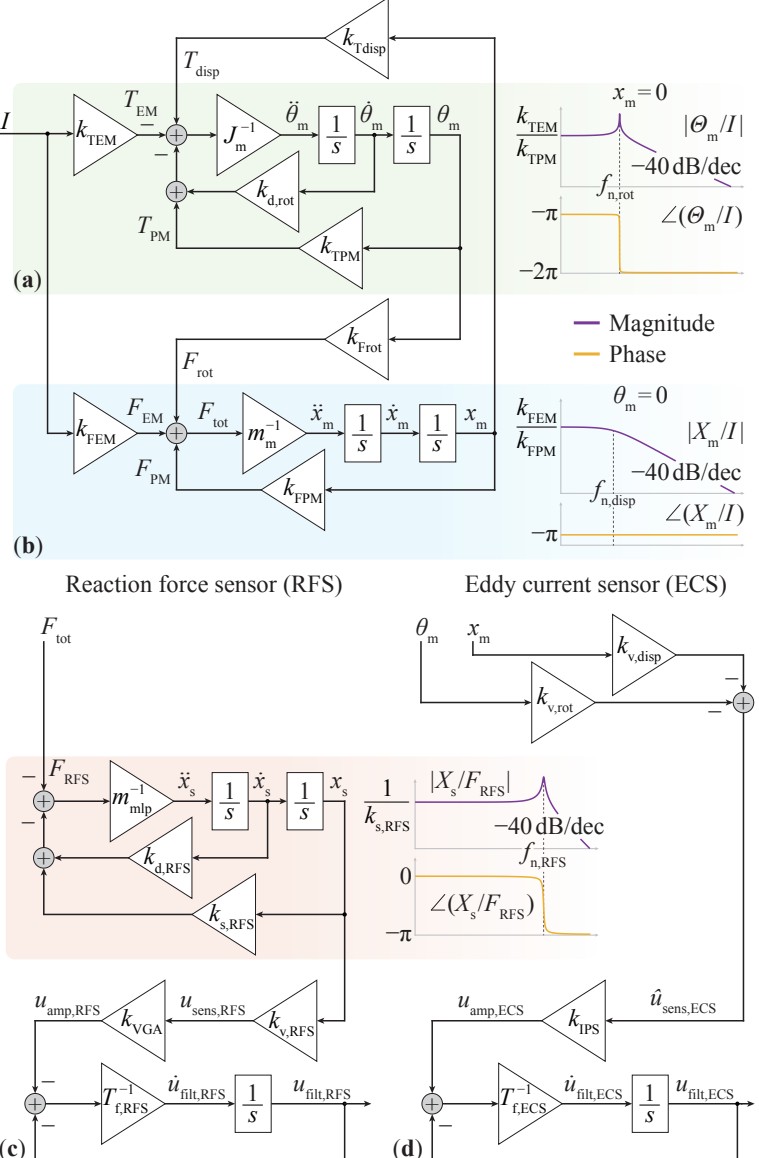

**Figure 5.** Block diagram of the mover's dynamics for (**a**) tilting around the radial axes and (**b**) radial displacement, where the different dynamics are also represented as Bode diagrams. Here, only the displacement in the $x$ direction and the related tilting angle around the $y$-axis, denoted as $\theta_{\text{m}}$, are considered. Due to symmetry, the same model holds for the $y$ displacement and the tilting angle around the $x$-axis. (**c**) Block diagram of the RFS system dynamics with the second-order system modeling the movement of the force sensor's sensing side, followed by the mechanical-to-electrical conversion of the strain gauges, amplification, and first-order active electronic filter. For details on the electrical diagram of the RFS, refer to [12]. (**d**) Block diagram of the ECS with the zero-order model describing the conversion from the mover's position to demodulated voltage, followed by the amplification and the first-order passive electronic filter.

For the ECS, the high-speed dynamics of the IPS2550 are simplified with the adjustable gain $k_{\mathrm{IPS}}$ (see Figure 5d and Table 1). To ensure a balanced comparison, the first-order analog low-pass filter is designed to mimic the phase response of the force sensor, complemented by a corresponding filter. Despite the difference in the orders of the two sensing systems, their phase responses are harmonized up to 20 Hz by calibrating the cutoff frequency of the ECS to $f_{\mathrm{f,ECS}} = 1/(2\pi \cdot T_{\mathrm{f,ECS}}) = 35$ Hz. Nonetheless, a deviation up to 1 dB at 20 Hz in the normalized magnitude responses is observed, attributable to the resonance peak in the force sensor's TF.

In evaluating the modeling aspects of the two sensing systems, it is noted that the RFS exhibits a higher level of complexity due to its second-order TF behavior. Conversely, the ECS, characterized as a zero-order system, presents a more straightforward modeling process. However, this simplicity is accompanied by practical limitations, notably the constraint against encasing the mover in an electrically conductive capsule. Such a drawback must be weighed against the ease of modeling the ECS offers, highlighting the necessity of a balanced assessment when considering integrating these sensing systems into the MLP.

### 4.2. Dynamic Model Verification and Calibration

For verifying the dynamic model of the MLP using the ECS, two measurements of the TF injecting sinusoidal currents $I$ in the EMs and observing the sensor's output $u_{\mathrm{out}}$ (defined as the sum of the filtered voltage $u_{\mathrm{filt}}$ of Figure 5 and the eventual disturbance voltage $u_{\mathrm{dist}}$) are performed, following the procedure described in [12]. The measured system's TF $G_{\mathrm{meas,x}} = U_{\mathrm{out,x}}/I_{\mathrm{x}}$ with the mover levitating, free to move only along the $x$-axis, and free to tilt around the $y$-axis, is shown in Figure 6a as a series of cyan points in the frequency range 0.1 Hz–200 Hz. Compared to the theoretical TF $G_{\mathrm{tot,x}} = U_{\mathrm{filt,x}}/I_{\mathrm{x}}$ (solid red line), the same trend of the theoretical model of Figure 5 is visible in the measurement, namely, the static gain and the peak due to the mover's tilting around 2 Hz. However, a significant mismatch starting from 2 Hz must be corrected with a second measurement (calibration) where the mover is removed from the MLP and the TF measurement is repeated by injecting currents in the EMs, as described in [12]. Following the model of Figure 5, the ECS should register zero voltages $u_{\mathrm{filt,x}}$ and $u_{\mathrm{filt,y}}$, as the mover's virtual position is zero. However, as shown in Figure 6b, TFs with a noticeable gain are measured, indicating parasitic couplings in the system that must be compensated. $G_{\mathrm{dist,xx}}$ represents the disturbance for the modeled TF $G_{\mathrm{tot,x}}$ and is obtained by injecting $I_{\mathrm{x}}$ in the EMs and measuring $u_{\mathrm{out,x}}$ without the mover. Adding the fitted frequency response of $G_{\mathrm{dist,xx}}$ to the modeled TF $G_{\mathrm{tot,x}}$, the measurement performed with the mover levitating is reproduced with better accuracy (cf. $G_{\mathrm{meas,x}}$ with $G_{\mathrm{tot,x}} + G_{\mathrm{dist,xx}}$ in Figure 6a). Another disturbance, which is measured by reading the sensor output $u_{\mathrm{out,x}}$ while the system is excited with the current $I_{\mathrm{y}}$ and without the mover, is the cross-coupling between the axes, indicated as $G_{\mathrm{dist,yx}}$ in Figure 6b. This disturbance is neglected since the gain is about 160 times lower than the measured system's response $G_{\mathrm{meas,x}}$ in the lower frequency range.

Accordingly, the observer (Kalman filter) for the ECS that extracts the mover's position $(x_{\mathrm{m}}, y_{\mathrm{m}})$ and tilting angles around the $x$- and $y$-axes is implemented considering only the calibrated model $G_{\mathrm{tot,x}} + G_{\mathrm{dist,xx}}$. In contrast, the Kalman filter for the RFS contains the supplemental cross-coupling correction terms (see [12]). In fact, the gain of $G_{\mathrm{dist,yx}}$ in Figure 6b is not negligible compared to the calibrated model $G_{\mathrm{tot,x}} + G_{\mathrm{dist,xx}}$ shown in Figure 6a.

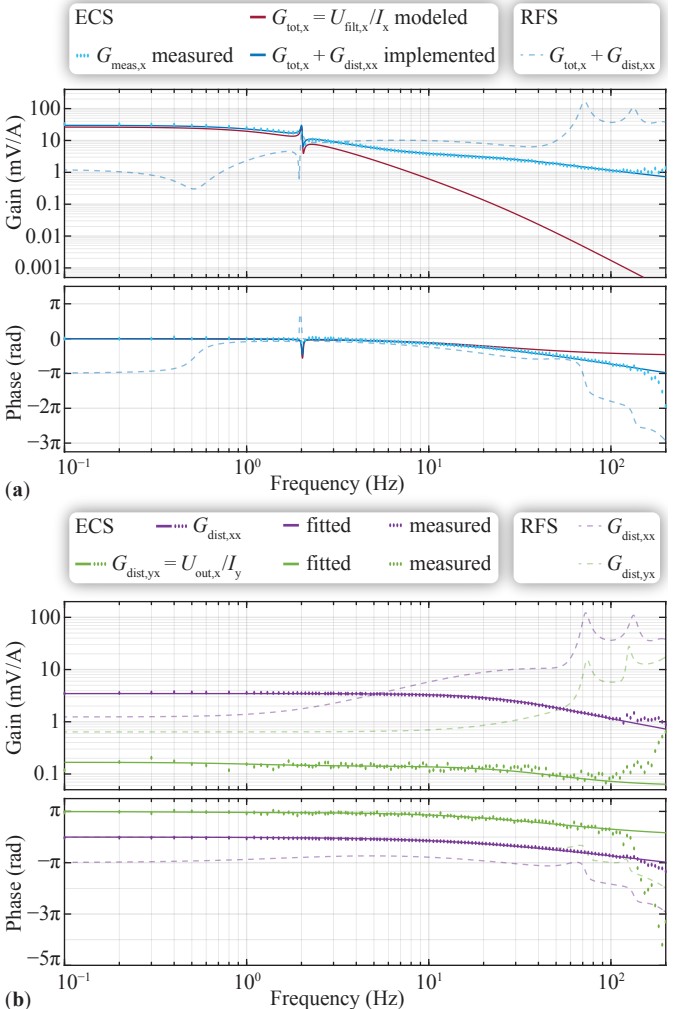

(a)

(b)

**Figure 6.** TFs measurements to prove and calibrate the model presented in Figure 5. (**a**) shows the frequency responses of the model, measurement, and observer implementation for the *x*-axis while the mover levitates from an input current in the *x* direction to the ECS's output. (**b**) shows the model disturbance $G_{\text{dist},xx}$ and the cross-coupling between the *y* current and the *x* sensor output $G_{\text{dist},yx}$ without mover for the ECS. The implemented TF, disturbance TF, and cross-coupling TF of the RFS found in [12] are overlapped with dashed curves. The measured phase lies within $[-\pi, \pi]$, but has been unwrapped for the representation (i.e., adjusted by adding or subtracting $2\pi$ to targeted phase values to ensure a continuous and smooth representation without discontinuities).

## 5. Comparative Results

This chapter compares the performance between the ECS and the RFS introduced in [12] and briefly described in Section 3 for steady-state levitation of the mover, which has been achieved using both sensing methods. A thorough analysis and comprehensive measurements have been conducted to understand the differences in performance and note the variations in the hardware realizations of the sensors.

### 5.1. Steady-State Levitation Comparison

The radial coordinates' reference is set to zero ($x_{\text{m}}^* = y_{\text{m}}^* = 0$) for the steady-state levitation comparison between the ECS and RFS, and a zero calibration of both sensors is performed without the mover. In this configuration, there are no reaction forces (RFS) or induced voltages (ECS) acting on the sensor. Afterward, the mover is brought by hand above the stator, and during free levitation, the controller keeps the mover at the previously calibrated point of zero reaction forces (RFS) or zero induced voltages (ECS). The results

shown in Figure 7a indicate that the estimated mover's position using the ECS is close to the reference, with small fluctuations within the recorded time interval of twenty seconds. Conversely, when the RFS is used, the mover's displacements from the centered position are larger, caused by the force sensor's drift over time, which is compensated with an outer current controller loop (see [12]). Despite all the disturbances and noise the RFS is facing in this application (see [12]), we manage to have a stable levitation with deviations of up to 5.5 mm, which is 5.3 % of the levitation height.

The different performances resulting for the two sensors are also visible in 2D plots without direct time dependency shown in Figure 7b–e, where the $x, y$ quantities (observed mover's position, observed mover's angle, measured sensor's voltage, and measured control current) are plotted over the recorded levitating period. The standard deviation (STD), i.e., the measure of the amount of variation of the mover's radial position relative to its expected mean value, is calculated from the recorded data over the entire period as a performance metric. The STD of the mover's position for the RFS is $(\sigma(x_{\mathrm{m}}), \sigma(y_{\mathrm{m}}))_{\mathrm{RFS}} = (1.26\,\mathrm{mm}, 1.60\,\mathrm{mm})$, whereas the ECS performs better with the STD of $(\sigma(x_{\mathrm{m}}), \sigma(y_{\mathrm{m}}))_{\mathrm{ECS}} = (0.05\,\mathrm{mm}, 0.15\,\mathrm{mm})$. This result is confirmed considering Figure 7b, where larger mover's displacements are visible using the RFS, and relatively smaller deviations are observed when using the ECS.

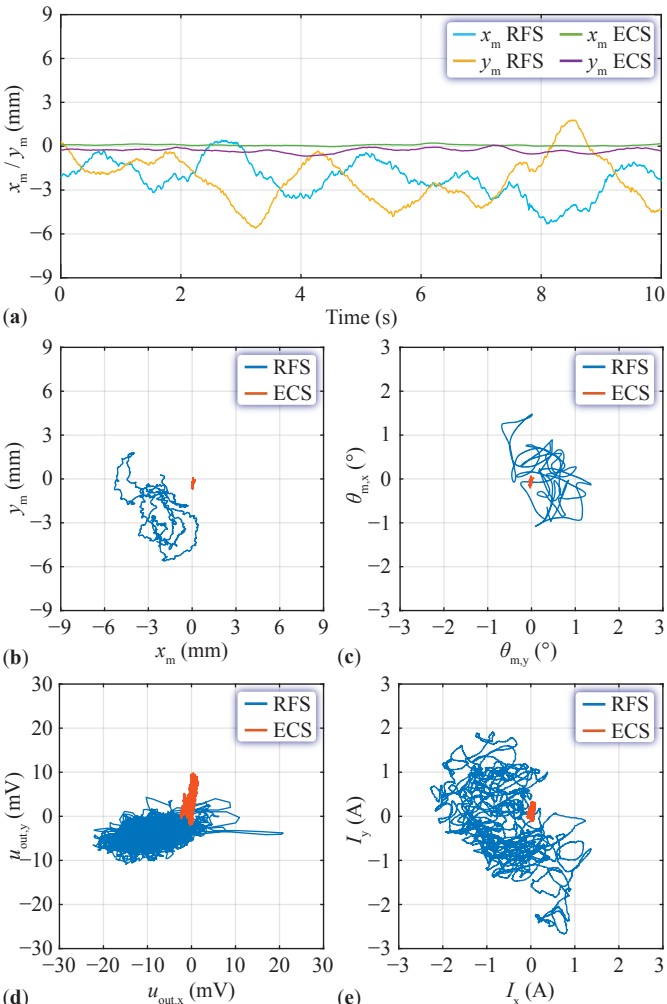

**Figure 7.** Comparison between ECS and RFS during steady-state levitation of the mover with a reference position $x_{\mathrm{m}} = y_{\mathrm{m}} = 0$. (**a**) shows the time-domain curves of the observed mover's position, also reported as a 2D plot in (**b**). The recording time is 10 s, with 2.5 kHz sampling frequency resulting in a total of 25,000 data points. (**c**–**e**) show the observed mover's tilting angle around the $x, y$-axes, the measured position sensors' output voltage, and the measured control currents, respectively. The measurement data clearly indicate better ECS position measurement performance.

Considering the plots in Figure 7b–e and focusing on the RFS's traces, it can be seen that the control current has a similar shape as the inverse of the mover's position. In other words, a rotation by 180° around the center is required for the plot displaying the mover's position to see that the shape is correlated to the current. The plot rotation corresponds to a negation of the quantities $(x_\mathrm{m}, y_\mathrm{m})$ since a positive current is required to counteract a negative mover's position, as depicted in Figure 2a and in the block diagram of the MLP in Figure 5b. The mover's angle adapts depending on the mover's position, e.g., the tilting angle around the $y$-axis, $\theta_\mathrm{m,y}$, varies with a radial displacement along the $x$-axis with $y_\mathrm{m} = 0$, explaining the direct correlation between the two plots (cf. Figure 7b with c, and note that $\theta_\mathrm{m,y}$ is shown on the horizontal axis as $x_\mathrm{m}$). Moreover, the controller cannot control the mover's angle to a reference value, but only actively dampens oscillations of the tilting angle around the $x, y$-axes occurring at the natural frequency $f_\mathrm{n,rot} = 2\,\mathrm{Hz}$ (see Table 3 and [12]). A similar correlation must exist between the sensed voltage and the observed mover's position and angle, as the latter quantities are extracted from the sensor's signal. However, it is difficult to identify a similar shape between the force sensor's voltage and the observed mover's position and tilting angle, indicating that disturbances and/or noise are present in the sensed signal. This statement is related to the TFs seen in Figure 6a and b, where the gain in the measured disturbance $G_\mathrm{dist,xx}$ and cross-coupling $G_\mathrm{dist,yx}$ is similar to the gain in the measured system's TF (matched by $G_\mathrm{tot,x} + G_\mathrm{dist,xx}$ in [12]). Due to lower SNR, the ECS trajectories presented in Figure 7 appear relatively small; a visual correlation between the measured quantities becomes evident in the presentation of results in Section 6.

### 5.2. Analysis of Performance Variance

This section highlights additional distinctions between the ECS and RFS-based MLP systems to explain their different performances. The first difference relates to a stronger correlation between the implemented and measured TFs observed for the ECS, as shown in Figure 6a. The observer implementation for the RFS system described in [12] differs from the measurements in the lower frequency range from 0.1 Hz to 0.7 Hz. It should be noted that the presented solution performs the best among several fine-tuning attempts to improve the match in that frequency range by adjusting the constants shown in the block diagram of Figure 5a,b (i.e., changing the shape of the modeled TF).

The second difference relates to a comparative time-domain measurement without the mover and the results are shown in Figure 8. Individually, both sensors are calibrated to zero at time 0 s using averaged data from past measurements while applying zero current to the EMs and continuously switching the 40 V DC-link voltage of the full-bridge two-level EMs' drive inverter. Subsequently, the position and current sensor signals are recorded for one second. The amplitude of the noise voltage from the force sensor is larger than that from the eddy current sensor, as shown in Figure 8a as a time-domain plot and Figure 8c as a 2D plot. In quantitative terms, the RMS noise amplitude for the force sensor is $(\hat{u}_\mathrm{out,x}, \hat{u}_\mathrm{out,y})_\mathrm{RFS} = (1.87\,\mathrm{mV_{rms}}, 1.70\,\mathrm{mV_{rms}})$, and for the eddy current sensor, it is $(\hat{u}_\mathrm{out,x}, \hat{u}_\mathrm{out,y})_\mathrm{ECS} = (0.22\,\mathrm{mV_{rms}}, 0.31\,\mathrm{mV_{rms}})$. Furthermore, the same noise on the current signal is measured since the same inverter is used for testing both sensing systems, as illustrated in Figure 8b,d. For completeness, the corresponding RMS amplitudes are $(\hat{I}_\mathrm{x}, \hat{I}_\mathrm{y})_\mathrm{RFS} = (13.5\,\mathrm{mA_{rms}}, 8.2\,\mathrm{mA_{rms}})$ and $(\hat{I}_\mathrm{x}, \hat{I}_\mathrm{y})_\mathrm{ECS} = (13.4\,\mathrm{mA_{rms}}, 8.1\,\mathrm{mA_{rms}})$. The Kalman filter includes all these measurement uncertainties by considering the voltage and current noise variance to adjust the Kalman gain when estimating the mover states. Therefore, due to the different noise levels in the position sensors, the estimating performance of the Kalman filter is different.

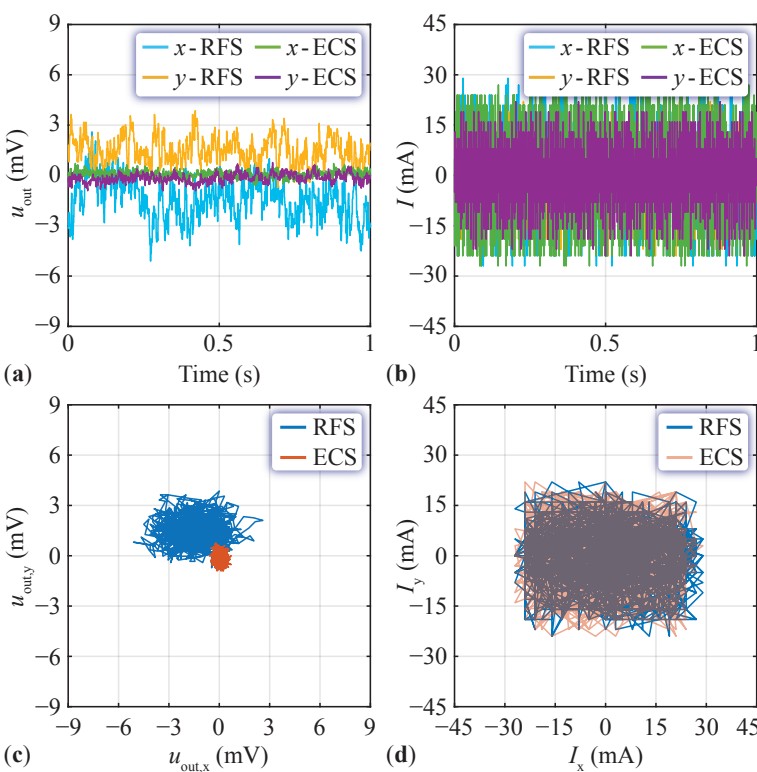

**Figure 8.** Comparison of static measurements performed without mover, where the signals from the position sensors (**a**,**c**) and EM current sensors (**b**,**d**) are recorded for one second, indicating that the output of the RFS has a higher noise level than the ECS.

The third difference is the signal-to-noise ratio (SNR), which is calculated to investigate both sensing systems by considering the waveforms in Figure 8a. The calculation assumes a displacement of the mover in the radial direction, neglects the tilting around the $x, y$-axes ($\theta_{\text{m,y}} = \theta_{\text{m,x}} = 0$), and takes no control action ($I = 0$). According to the model presented in Figure 5, the static gain from the mover's position to the sensors' output is calculated and measured as

$$\frac{u_{\text{filt,ECS,x}}}{x_{\text{m}}} = k_{\text{v,disp}} \cdot k_{\text{IPS}} = 12.3\,\text{mV/mm} \tag{2}$$

$$\frac{u_{\text{filt,ECS,y}}}{y_{\text{m}}} = k_{\text{v,disp}} \cdot k_{\text{IPS}} = 12.3\,\text{mV/mm} \tag{3}$$

$$\frac{u_{\text{filt,RFS,x}}}{x_{\text{m}}} = \frac{k_{\text{FPM}}}{k_{\text{s,RFS,x}}} \cdot k_{\text{v,RFS,x}} \cdot k_{\text{VGA}} = 6.3\,\text{mV/mm} \tag{4}$$

$$\frac{u_{\text{filt,RFS,y}}}{y_{\text{m}}} = \frac{k_{\text{FPM}}}{k_{\text{s,RFS,y}}} \cdot k_{\text{v,RFS,y}} \cdot k_{\text{VGA}} = 7.3\,\text{mV/mm}. \tag{5}$$

The difference between the RFS's $x, y$ quantities arises due to its asymmetrical construction (see [12] for details on the constants). Consequently, the SNR for a displacement of the mover by $x_{\text{m}} = y_{\text{m}} = 1\,\text{mm}_{\text{rms}}$ (RMS amplitude) around the radially centered position is given as

$$\text{SNR} = 20 \cdot \log_{10}\left(\frac{\hat{u}_{\text{filt}}}{\hat{u}_{\text{noise}}}\right), \tag{6}$$

where $\hat{u}_{\text{filt}}$ is the RMS amplitude of the sensed noise voltage due to the mover's displacement calculated with (2)–(5). $\hat{u}_{\text{noise}}$ is the RMS amplitude of the sensed voltage noise from Figure 8a. The resulting pairs are $(\text{SNR}_{\text{x}}, \text{SNR}_{\text{y}})_{\text{RFS}} = (10.5\,\text{dB}, 12.7\,\text{dB})$ and $(\text{SNR}_{\text{x}}, \text{SNR}_{\text{y}})_{\text{ECS}} = (35.0\,\text{dB}, 32.0\,\text{dB})$, indicating that the mover's displacement of $1\,\text{mm}_{\text{rms}}$ is distinguishable from noise for both sensors since positive values are obtained. However,

the ECS measures a cleaner signal because its SNR is higher by about 20 dB. From another perspective, the lower detectable limits for the mover's displacement are calculated by equating SNR = 0 and considering the measured noise amplitude. The results are $(x_{m,min}, y_{m,min})_{RFS} = (0.30\,\text{mm}_{rms}, 0.23\,\text{mm}_{rms})$ and $(x_{m,min}, y_{m,min})_{ECS} = (0.018\,\text{mm}_{rms}, 0.025\,\text{mm}_{rms})$, indicating that the measurement with the ECS is about ten times higher in resolution. The difference is mainly due to the different noise level since the conversion from the mover's position to voltage differs by about a factor of two for the ECS and the RFS (cf. (2) and (3) with (4) and (5)).

The difference in the sensors' noise levels arises from various factors. First, the amplifier for the RFS is custom-made. It consists of a buffered ultra-low noise voltage reference [23] to excite the strain gauges, a manual offset compensation circuit for each axis, a variable gain amplifier (VGA) combined with an active *RC* filter for each axis, and a four-channel 16-bit analog-to-digital converter (ADC) [24]. Instead, the coil excitation for the ECS and the processing of the received signal (amplification and demodulation) occur in a single integrated circuit [20]. The hardware is completed with a passive *RC* filter and the same ADC as the force sensor's one. Therefore, fewer components can pick up noise in the ECS's hardware. Second, the operating frequency range and the total gain for the electrical signal of the two sensors are substantially different. The amplifier for the RFS operates from zero frequency up to the bandwidth of the VGA (18 MHz) with a gain of 10 V/mV (80 dB). In contrast, the amplifier for the ECS amplifies signals by 240 V/V (47.6 dB) only at the excitation frequency of 3.1 MHz since the subsequent demodulation rejects noise at other frequencies. Hence, the RFS's amplifier enhances the probability of noise amplification. Finally, the spatial placement of the amplifier circuits within the MLP impacts the noise at their outputs. The sensing elements of both sensors (aluminum body with strain gauges and electromagnetic coils) are placed in the neighborhood of the EMs, which inevitably radiate high-frequency (HF) electromagnetic fields due to the HF current components caused by the switched power converter. The RFS's amplifier is placed on the side of the assembly formed by the PM stator and EMs. In contrast, the ECS's amplifier is placed directly under the assembly, i.e., at the same location as the RFS's aluminum body (see Figures 1 and 4a). On the one hand, this arrangement can be advantageous for the RFS over the ECS because the source of electromagnetic radiation that can induce noise in the sensed signals is farther from the sensitive electronics. On the other hand, the analog signals must be routed over a longer distance for the RFS than for the ECS, rendering them more susceptible to noise. Thus, a definitive favorable placement cannot be determined in this context.

The main differences between the RFS and ECS are summarized in Table 2. A combination with another sensing principle could be employed for future developments on the RFS instead of using strain gauges that easily pick up noise and suffer from induced voltages. For example, precise eddy current or optical sensors [25] could sense the displacement of the RFS's sensing side due to the reaction forces from the mover. According to Table 2, the ECS is more robust against noise than the RFS for using an off-the-shelf force sensor. However, these approaches to advance the RFS increase the system's complexity and leave open challenges regarding the sensitivity and the cross-coupling between the axes, which must be solved by the mechanical design of the force sensor. When enhancing the force sensor's sensitivity through optimizing its mechanical design, reduced measurement bandwidth and lower load rating must be considered [26]. Conversely, it is observed that the sensitivity of the ECS does not negatively impact the measurement bandwidth. Enhancing the sensor's sensitivity through alternative coil designs or processing signals at higher frequencies does not necessarily result in a reduced bandwidth, as the IPS2550's processing delay discussed in Section 2.1 remains constant.

**Table 2.** Comparison of the main characteristics and performances of the RFS and the ECS. The standard deviation ($\sigma$) of the mover's position from the reference position $x_{\text{m}}^* = y_{\text{m}}^* = 0$ is calculated during steady-state levitation within a time interval of ten seconds. The RMS noise ($\hat{u}_{\text{out}}$) is measured at the sensors' output voltage without the mover levitating. The SNR is calculated using the RMS noise, (2)–(5), derived from the MLP's models given in Figure 5.

|  | RFS | ECS |
|---|---|---|
| Mover can be encapsulated in a stainless steel chamber | Yes | No |
| Trade-off between sensitivity and bandwidth | Yes | No |
| Electromagnetic and mechanical disturbances | Large | Small |
| Cross-coupling | Large | Negligible |
| $\sigma(x_{\text{m}})$ | 1.26 mm | 0.05 mm |
| $\sigma(y_{\text{m}})$ | 1.60 mm | 0.15 mm |
| $\hat{u}_{\text{out,x}}$ | 1.87 mV$_{\text{rms}}$ | 0.22 mV$_{\text{rms}}$ |
| $\hat{u}_{\text{out,y}}$ | 1.70 mV$_{\text{rms}}$ | 0.31 mV$_{\text{rms}}$ |
| SNR$_{\text{x}}$ | 10.5 dB | 35.0 dB |
| SNR$_{\text{y}}$ | 12.7 dB | 32.0 dB |

## 6. Additional Experimental Results for the Eddy Current Sensor

This chapter presents additional tests conducted with the ECS employed for position control to highlight features that can inspire new applications and show the MLP's servo capabilities that are not possible with the current version of the RFS.

### 6.1. Disturbance during Steady-State Levitation

In the first experiment, an external disturbance in the form of a hand-driven push on the mover during steady-state levitation allows for inspecting the levitation robustness of the MLP with ECS. As shown in Figure 9a, while the position and outer current controller are keeping the mover in the radially centered position, a disturbance is initiated at 5.6 s in the positive $y$ direction. The mover displaces 4.8 mm from the origin with an angle of $\theta_{\text{m,x}} = -1.9°$ and an initial speed of $dy_{\text{m}}/dt = 17.9$ mm/s. The approximate push force $F_{\text{push}}$ is calculated from the position and current curves using the force balance equation

$$m_{\text{m}} \cdot \frac{d^2 y_{\text{m}}}{dt^2} = F_{\text{push}} + F_{\text{PM}} + F_{\text{EM}} + F_{\text{rot}}$$
$$= F_{\text{push}} + k_{\text{FPM}} \cdot y_{\text{m}} + k_{\text{FEM}} \cdot I + k_{\text{Frot}} \cdot \theta_{\text{m,x}}. \tag{7}$$

The resulting impulse $F_{\text{push}} = 0.19$ N is counteracted by the controller, which steers the mover back to the origin in 0.8 s after the disturbance initiation with a speed of 14.9 mm/s.

As a consequence of the perturbation, the mover starts spinning around the $z$-axis. The spinning motion persists over time since this type of rotation cannot be stopped by magnetic torques or EMs. If the system is perfectly symmetric, the mover's rotation around the $z$-axis should not be reflected on the radial quantities, such as position and tilting angle. However, the effect of the axial rotation is visible on the radial axes as a sinusoidal waveform with a period of about 80 s in the time-domain plot of Figure 9a. This observation, coupled with occasional axial rotations around the $z$-axis during steady-state levitation, indicates that the manufactured MLP presents certain asymmetries.

Furthermore, the correlation between the current, the mover's position, and the tilting angle around the $x, y$-axes is visible in Figure 9b–e. The similarity between the mover's position and angle and the sensed voltage follows the model shown in Figure 5d. A positive mover radial position gives rise to a negative sensor voltage, and a positive tilting angle around the $x, y$-axes translates into a negative voltage.

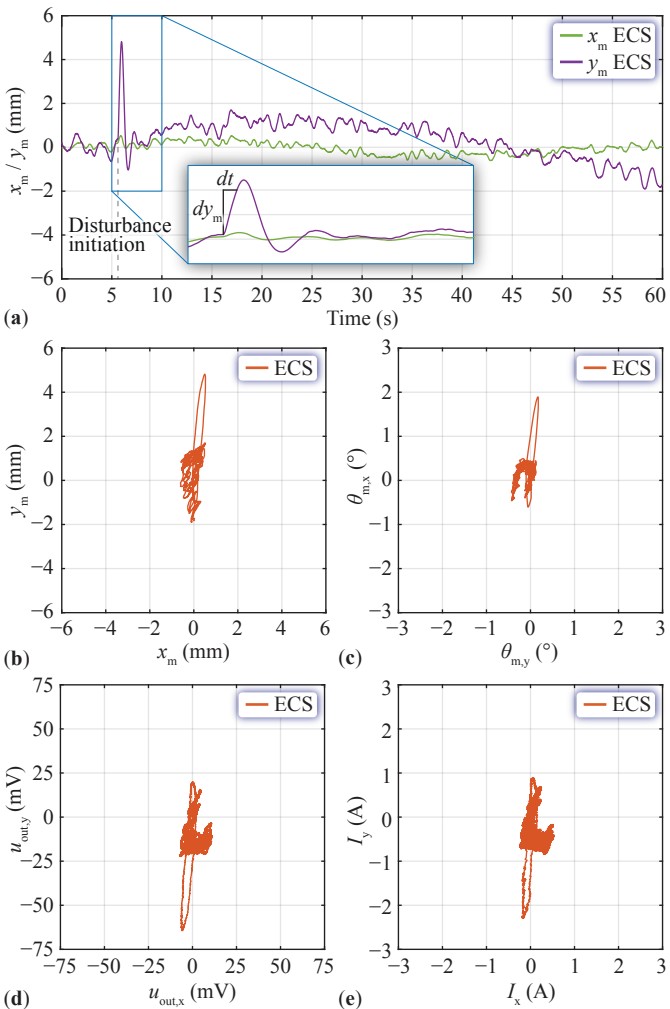

**Figure 9.** Experiment performed on the MLP with ECS position sensing, where a manual push is applied to the mover. (**a**,**b**) show the observed mover's position. (**c**–**e**) show the observed mover's tilting angle around the $x, y$-axes, the measured position sensor's output voltage, and the measured control current, respectively. The controller counteracts the disturbance; however, it could initiate a spinning motion of the mover around the $z$-axis, worsening the levitation performance.

### 6.2. Position Reference Tracking

In the second experiment, we fed the position controller with a time-varying position reference signal and observed the actual mover's position. The outer loop current controller (see Figure 8 in [12]) is omitted because it would inevitably interfere with the position controller. The references for the $x$- and $y$-axes position controllers are sinusoidal signals with an amplitude of 3 mm and a frequency of 0.1 Hz, and they are phase-shifted by 90° to steer the mover circularly within the $x, y$ plane. The measurements are depicted in Figure 10. The mover tracks the reference with an offset of $(x, y) = (-0.98\,\text{mm}, 0.60\,\text{mm})$ about the origin and a standard deviation of $(\sigma(x_\text{m} - x_\text{m,ref}), \sigma(y_\text{m} - y_\text{m,ref})) = (0.36\,\text{mm}, 0.24\,\text{mm})$ compared to the reference signals. The worse performance on the $x$-axis compared to the steady-state levitation is due to the asymmetry within the MLP that generates an offset and initiates the spinning of the mover around the $z$-axis. The spinning leads the mover toward positions that are harder to control, as shown between 90 s and 140 s in Figure 10a when $x_\text{m}$ is at the minimum. Moreover, it results in a distorted circular trajectory in the 2D plot of Figure 10b for negative $x_\text{m}$ and positive $y_\text{m}$.

Furthermore, the mover's tilting angle around the $x, y$-axes shown in Figure 10c assumes a value depending on the radial position, and it cannot be controlled to an arbitrary reference value with the available EMs since they can only steer the radial position.

However, active damping of the oscillations of the tilting angle around the $x, y$-axes is achieved because the controller only targets the specific frequency $f_{n,rot} = 2$ Hz.

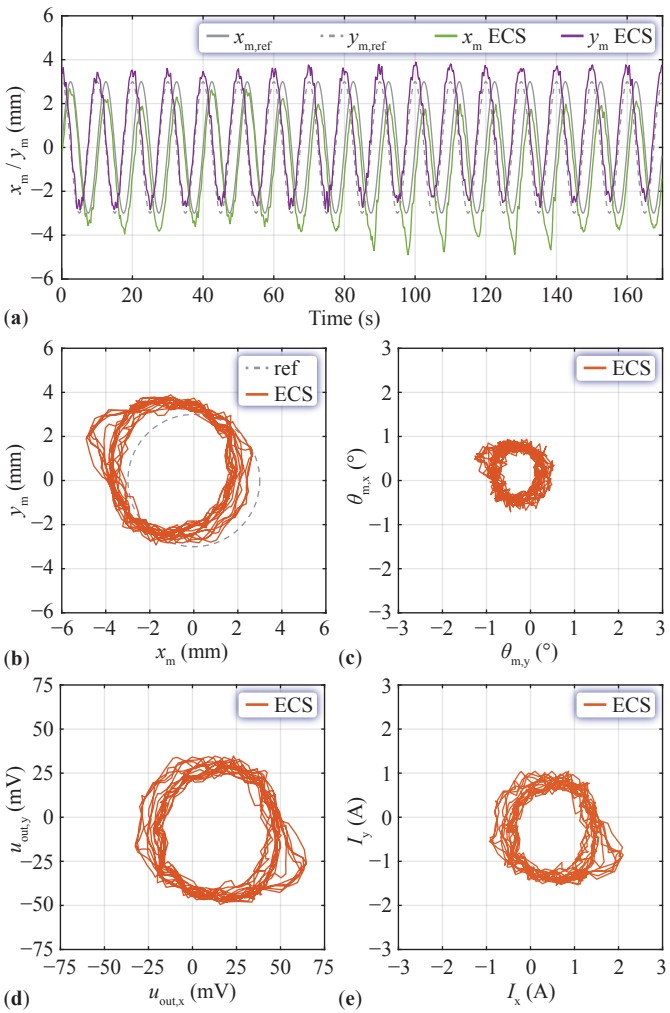

**Figure 10.** Dynamic experiment performed using the ECS, where the controller has to steer the mover along a circular reference in the $x, y$ plane. (**a,b**) show the observed mover's position. (**c–e**) show the observed mover's tilting angle around the $x, y$-axes, the measured position sensor's output voltage, and the measured control current, respectively. An asymmetry in the MLP gives rise to an offset position and a spinning of the mover around the $z$-axis that negatively affect the tracking performance.

### 6.3. Position Reference Tracking with Payload

The last presented test consists of loading the mover with an additional weight and steering it in the $x, y$ plane with a circular position reference to simulate the dynamical positioning of an actuator or a payload. The levitation height decreases to 70 mm due to the additional weight of 0.38 kg on the mover. Due to the payload, new constants in the model of Figure 5a,b, and d are required, leading to the values listed in Table 3. With this new configuration, the gain of the ECS $k_{IPS}$ is unchanged since the voltage bounds of the ADC are not exceeded for the controllable range of the mover's positions. The natural frequencies of the mover's dynamics increase to $f_{n,disp} = 1.74$ Hz and $f_{n,rot} = 2.62$ Hz due to a greater increase in magnetic stiffness compared to the mass and moment of inertia. Nevertheless, the low-pass filter's time constant (or cutoff frequency of 35 Hz) described by $T_{f,ECS}$ is unchanged since it is already adequate, i.e., it is ten times larger than the mover's natural frequencies. New calibrations are not required since they are performed without the mover; thus, the same transfer functions as shown in Figure 6d are implemented in the observer. The recorded data are displayed in Figure 11, where the amplitude and the

frequency of the position reference signals are set to 2 mm and 0.1 Hz, respectively. The tracking performance is measured with the standard deviation from the reference signals, resulting in $(\sigma(x_m - x_{m,ref}), \sigma(y_m - y_{m,ref})) = (0.40\,\text{mm}, 0.70\,\text{mm})$. A large part of the error is attributed to the offset from the center $(x, y) = (-0.33\,\text{mm}, -0.27\,\text{mm})$ arising from the asymmetry in the MLP. The spinning of the mover around the $z$-axis is inhibited by a manual (contactless) control during this measurement. Otherwise, the asymmetry combined with the spinning would destabilize the mover. For the manual control, an additional magnet is glued on the payload, and another magnet is brought into its vicinity to generate a compensating torque whenever a rotation is initiated. In a final design, this very basic technique would have to be replaced by an additional autonomous system consisting of non-axially symmetric magnets placed on the mover, a sensor, and electromagnets, which, however, increase the dimensions and complexity of the MLP.

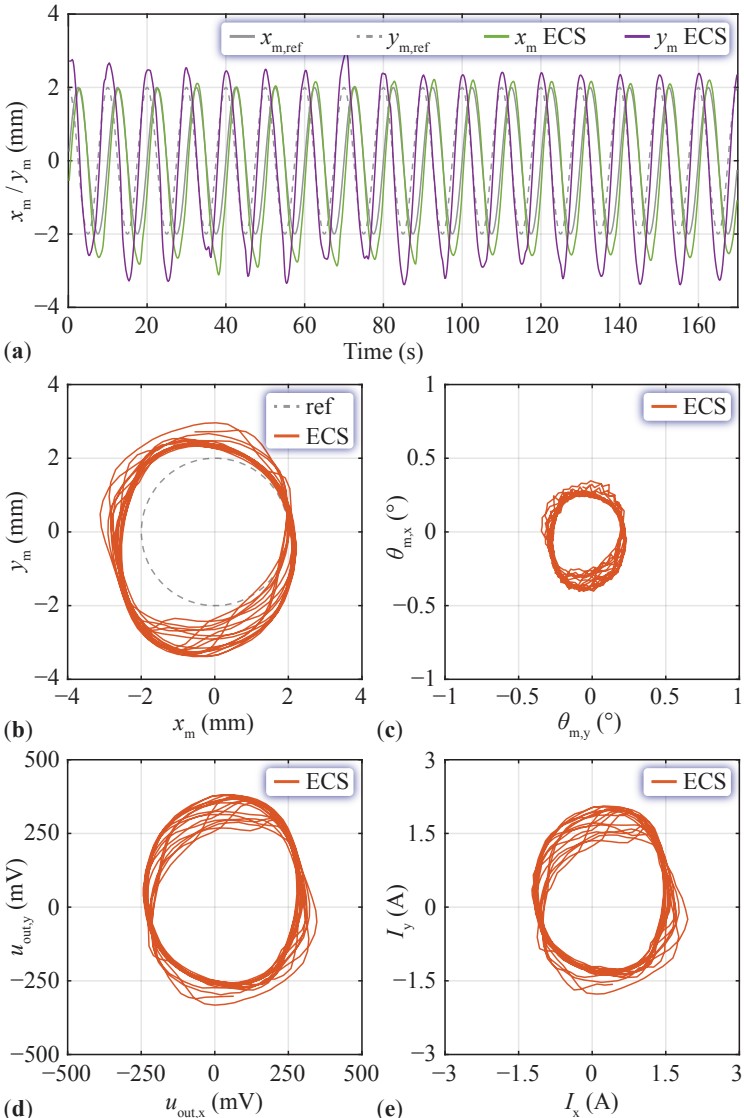

**Figure 11.** Dynamic experiment with the ECS similar to that shown in Figure 10, but with an additional weight of 0.38 kg placed on the mover that reduces the levitation height to 70 mm and dictates a change in the model parameters. (**a**,**b**) show the observed mover's position. (**c**–**e**) show the observed mover's tilting angle around the $x, y$-axes, the measured position sensor's output voltage, and the measured control current, respectively. An asymmetry of the PMs in the MLP and the load causes a positional offset, impairing the tracking performance.

**Table 3.** MLP and ECS parameters with and without payload presence on the mover. The mover's mass and the payload are included in $m_\mathrm{m}$. The only ECS parameters that differ from Table 1 are $k_\mathrm{v,disp}$ and $k_\mathrm{v,rot}$.

| Magnetic levitation platform without payload | | |
|---|---|---|
| Levitation height | $h$ | 104 mm |
| Characteristic dimension | CD | 207 mm |
| Mover weight | $m_\mathrm{m}$ | 0.36 kg |
| Mover moment of inertia | $J_\mathrm{m}$ | 0.58 gm² |
| Radial stiffness | $k_\mathrm{FPM}$ | 32.8 N/m |
| Displacement torque const. | $k_\mathrm{Tdisp}$ | 0.25 Nm/m |
| Rotational stiffness | $k_\mathrm{TPM}$ | 1.6 mNm/° |
| Rotational force const. | $k_\mathrm{Frot}$ | 4.4 mNm/° |
| EMs force const. | $k_\mathrm{FEM}$ | 65 mN/A |
| EMs torque const. | $k_\mathrm{TEM}$ | 0.93 mNm/A |
| Rotational damping | $k_\mathrm{d,rot}$ | 2 µNms/° |
| Rotation natural frequency | $f_\mathrm{n,rot}$ | 2 Hz |
| Radial disp. natural frequency | $f_\mathrm{n,disp}$ | 1.52 Hz |

| Magnetic levitation platform with payload of 0.38 kg | | | |
|---|---|---|---|
| $h$ | 70 mm | $m_\mathrm{m}$ | 0.74 kg |
| $k_\mathrm{FPM}$ | 88.1 N/m | $J_\mathrm{m}$ | 1.54 gm² |
| $k_\mathrm{TPM}$ | 7.3 mNm/° | $k_\mathrm{FEM}$ | 170.4 mN/A |
| $k_\mathrm{d,rot}$ | 20 µNms/° | $k_\mathrm{TEM}$ | 1.4 mNm/A |
| $f_\mathrm{n,rot}$ | 2.62 Hz | $f_\mathrm{n,disp}$ | 1.74 Hz |
| Eddy current sensor | | | |
| $k_\mathrm{v,disp}$ | 421.4 µV/mm | $k_\mathrm{v,rot}$ | 250 µV/° |

## 7. Conclusions

This paper comparatively evaluates the performance of a reaction force-based position sensor (RFS) and an eddy current position sensor (ECS) in determining the radial position of the levitating permanent magnet (PM) mover in a magnetic levitation platform (MLP). The RFS measures the reaction forces on the stator due to the mover's motion, whereas the ECS operates high-frequency magnetic fields and measures the change in induced voltage in the ECS sensing coils placed on the stator due to the mover's motion. Both sensors were integrated with an observer-based controller approach, a necessity brought by the intrinsic dynamics inherent in each sensor, and exhibited commendable qualities for different fields of application. To ensure a fair comparison in terms of cost and sensor complexity, we chose an off-the-shelf force sensor for the RFS application. In this case, the ECS stood out for its superior precision, demonstrating an approximately ten times higher accuracy than the RFS during steady-state levitation. Its performance was evaluated by calculating the standard deviation from the reference mover's position, resulting in $(\sigma(x_\mathrm{m}), \sigma(y_\mathrm{m}))_\mathrm{ECS} = (0.05\,\mathrm{mm}, 0.15\,\mathrm{mm})$ for a levitation height of $h = 104\,\mathrm{mm}$. It also displayed remarkable resilience to disturbances, noise, and cross-couplings, thus offering a 20 dB higher signal-to-noise ratio $(\mathrm{SNR_x}, \mathrm{SNR_y})_\mathrm{ECS} = (35.0\,\mathrm{dB}, 32.0\,\mathrm{dB})$ and enhanced performance reliability compared to the off-the-shelf RFS. On the other hand, the RFS demonstrates a unique advantage in scenarios where conductive materials are present in the air gap, a condition that limits the applicability of the ECS. This makes the RFS a valuable alternative for specialized applications, ensuring its relevance and utility in the diverse operational contexts of MLPs. Furthermore, the ECS showcased remarkable capabilities in handling static and dynamic tracking performances despite the challenges posed by asymmetries due to manufacturing tolerances in the MLP, highlighting its adaptability and robustness in various operational circumstances. In summary, while both sensors bring valuable qualities, the ECS distinguishes itself with higher precision and a higher SNR, making it exceptionally proficient in most MLP applications. The RFS, however, maintains its significance by offering applicability in the presence of conductive obstructions within the air gap, ensuring its indispensable role in specialized MLP scenarios.

**Author Contributions:** Conceptualization, R.B.; methodology, R.B. and S.M.; software, R.B.; validation, R.B.; formal analysis, R.B. and S.M.; investigation, R.B.; resources, J.W.K.; data curation, R.B.; writing—original draft preparation, R.B. and S.M.; writing—review and editing, R.B., S.M. and J.W.K.; visualization, R.B., S.M. and J.W.K.; supervision, J.W.K.; project administration, J.W.K.; funding acquisition, J.W.K. All authors have read and agreed to the published version of the manuscript.

**Funding:** This research was funded by the Else und Friedrich Hugel-Fonds für Mechatronik/ETH Foundation.

**Data Availability Statement:** Data presented in this study are available on request from the corresponding author.

**Acknowledgments:** The authors are very much indebted to the "Else und Friedrich Hugel-Fonds für Mechatronik/ETH Foundation", which generously supports the research on magnetic levitation platforms with extreme levitation distance at the Power Electronic Systems Laboratory of ETH Zurich.

**Conflicts of Interest:** The authors declare no conflicts of interest.

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
