# Peer review of "Comparative Analysis of Force and Eddy Current Position Sensing Approaches for a Magnetic Levitation Platform with an Exceptional Hovering Distance"

_actuators, doi:10.3390/act13040122_

Round 1
Reviewer 1 Report
Comments and Suggestions for Authors
This paper presented a comparative analysis of force and eddy current position sensing approaches for magnetic levitation platform with an exceptional hovering distance. The authors discussed the advantages and weaknesses of each sensor concept, explored operational principles and performance in levitation tests. The simulation and experiment are investigated. The eddy current sensor achieves a position tracking precision where the mover is vertically distanced at 104 mm from the stator. Generally, this paper was well organized and presented. It could be accepted to future publication if the authors can address some concerns as listed below:
(1) The magnetic levitation platform utilizes the permanent magnets to maintain the mover at the desired position in the axial direction and around the x and y axes. The radial (x, y direction) displacement is passively unstable due to magnetic forces that pull the mover away. Therefore, an active control of these two DOFs is strictly necessary. The sensors required for different control accuracy requirements are different. For example, two hall sensors can be used to detect displacement in the x and y directions to achieve stable suspension. Why did the author choose eddy current displacement sensors? What are the performance requirements for the studied maglev platform? The author needs to elaborate.
(2) This paper comparatively evaluates the performance of a reaction force-based position sensor and an eddy current position sensor in determining the radial position of the levitating permanent magnet mover in a magnetic levitation platform. The working principles of both need to be introduced. Please consider it and state them.
(3) Some drawings should be clear. The conclusion is not enough, please refining them.
Author Response
Dear reviewer,
Thank you very much for your review and your helpful comments.
(1) Thank you for raising these points.
Hall effect sensors are not applicable for such a magnetic levitation platform because of the substantial decay of the magnetic field strength with increasing distance. A significant amplification of the sensed signal would be required for large air gaps to detect a change in magnetic field due to a displacement of the mover in the x, y direction. However, high gain is not applicable as it would lead to saturation of the measuring circuit output due to the high magnetic field near the stator permanent magnet. Written in the introduction of [Bonetti et al., Reaction Force-Based Position Sensing for ...].
Instead, the gain of the presented eddy current sensor can be large since the sensed electrical voltage is only influenced by the mover's motion.
We addressed your comment in the introduction.
Regarding the performance, we present a research demonstrator and we are evaluating "how" and "how well" the position can be controlled., rather than fulfilling a given specification.
(2) Thank you for this comment about the conclusion.
We added the principles of both sensors in the conclusion.
(3) We have checked all figures for clarity and reviewed the conclusions.
Reviewer 2 Report
Comments and Suggestions for Authors
This paper gives a detailed comparisons between the eddy current position sensor and the force-based position sensing based on a magnetic levitation platform. The paper is good structured and well written with abundant denotations, simulations, calibrations and verifications. And the sensing scenery when conductive materials lie in the sensor and target objects do exists where the ECS would be unfunctional. This comparative study is meaningful to the future study and application for the related fields when distant position sensing is necessary.
In general, this paper is suggested to be published.
Author Response
Dear reviewer,
Thank you very much for your review and your kind words on the value of the paper.
Reviewer 3 Report
Comments and Suggestions for Authors
Comments to the authors:
Page 1: line 40: Athors should insert: ”Magnetic levitation and precise position measurement is also important when using highly sensitive temperature-compensated quartz sensors that measure position based on capacitive change, as shown in ref.:”
-Detection principles of temperature compensated oscillators with reactance influence on piezoelectric resonator. Sensors. 2020, vol. 20, iss. 3, p. 1-18. ISSN 1424-8220. https://www.mdpi.com/1424-8220/20/3/802
-Temperature-compensated capacitance-frequency converter with high resolution. Sensors and actuators. A, Physical, ISSN 0924-4247, 2014, vol. 220, p. 262-269, doi: 10.1016/j.sna.2014.09.022.
Authors should include mentioned references in the text.
Page 7: line 201: Is there any mark on the force sensor?
Page 9: Figure 5: The picture is not clear enough. There should also be an electrical diagram for a clearer presentation of the operation. It is also unclear what is marked with b) in the picture.
Page 12: Figure 7: How was the position measured along the x and y axes at a height of 104 mm? How was the reference static position determined? How does the temperature affect the stability of the position, given that the excitation coil is made of copper?
Page 15: How many bits of ADC did you use, because the signal-to-noise ratio when measuring the position also depends on it?
Author Response
Dear reviewer,
Thank you very much for your review and your helpful comments.
(Page 1) Thank you for raising this point.
We added the suggested sentence and references in a more suitable paragraph than the one specified.
(Page 7) Thank you for this comment.
We added the type and cited the datasheet of the used force sensor.
(Page 9: Figure 5) Thank you for asking a clarification.
The electrical diagram of the RFS can be found in [Bonetti et al., Reaction Force-Based Position Sensing for ...], and for that we added the citation in the caption of Figure 5. The electrical diagram of the ECS is as simple as shown in the block diagram since most of the signal processing takes place in a single integrated circuit.
We moved a) and b) to the colored area to clearly mark what they are referring to.
(Page 12: Figure 7) Thank you for raising these questions.
The shown x,y position is the position determined by the observer (specified in the caption as "observed mover's position").
Both sensors have been calibrated to the zero point without the mover. n this configuration, there are no reaction forces (RFS) or induced voltages (ECS) acting on the sensor. Afterward, the mover is brought by hand above the stator, and during free levitation, the controller tries to keep the mover at the previously calibrated point of zero reaction forces (RFS) or zero induced voltages (ECS). We added this explanation in subsection 5.1.
We did not conduct a study on the stability of the position over temperature. However, the change of resistance due to heating does not play a role since the integrated circuit (IPS2550) drives the excitation coil at the self resonant frequency, which is unaffected by temperature. The self inductance and parasitic capacitance only depend on the size/position of the coil.
(Page 15) Thank you for asking this detail.
Both sensors use the same ADC, see line 395. Since we compare both signal-to-noise ratios using a common ADC, the SNR of the ADC is equal in both cases.
We added the bit count (16-bit) where the ADC is mentioned.